# A Multi-Trait Gaussian Kernel Genomic Prediction Model under Three Tunning Strategies

**DOI:** 10.3390/genes13122279

**Published:** 2022-12-03

**Authors:** Abelardo Montesinos-López, Bernabe Cano-Páez, J. Cricelio Montesinos-López, Moisés Chavira-Flores, Osval A. Montesinos-López, José Crossa

**Affiliations:** 1Statistics Study Program, Universitas Negeri Yogyakarta, Yogyakarta 55281, Indonesia; 2Centro Universitario de Ciencias Exactas e Ingenierías (CUCEI), Universidad de Guadalajara, Guadalajara 44430, Jalisco, Mexico; 3Facultad de Ciencias, Universidad Nacional Autónoma de México (UNAM), México City 04510, Mexico; 4Department of Public Health Sciences, University of California Davis, Davis, CA 95616, USA; 5Instituto de Investigaciones en Matemáticas Aplicadas y Sistemas (IIMAS), Universidad Nacional Autónoma de México (UNAM), México City 04510, Mexico; 6Facultad de Telemática, Universidad de Colima, Colima 28040, Colima, Mexico; 7International Maize and Wheat Improvement Center (CIMMYT), Km 45, Carretera Mexico, Veracruz 52640, Edo. de México, Mexico; 8Colegio de Postgraduados, Montecillos 56230, Edo. de México, Mexico

**Keywords:** kernels, multi-trait, Bayesian optimization, grid search, genomic selection

## Abstract

While genomic selection (GS) began revolutionizing plant breeding when it was proposed around 20 years ago, its practical implementation is still challenging as many factors affect its accuracy. One such factor is the choice of the statistical machine learning method. For this reason, we explore the tuning process under a multi-trait framework using the Gaussian kernel with a multi-trait Bayesian Best Linear Unbiased Predictor (GBLUP) model. We explored three methods of tuning (manual, grid search and Bayesian optimization) using 5 real datasets of breeding programs. We found that using grid search and Bayesian optimization improve between 1.9 and 6.8% the prediction accuracy regarding of using manual tuning. While the improvement in prediction accuracy in some cases can be marginal, it is very important to carry out the tuning process carefully to improve the accuracy of the GS methodology, even though this entails greater computational resources.

## 1. Introduction

Genomic selection (GS) is frequently used for genetic improvement and has many advantages over phenotype-based selection [1]. Nevertheless, breeders face an adversity of challenges to improve the accuracy of the GS methodology, similar to multi-trait (MT) genomic prediction models, which take advantage of correlated traits to improve prediction accuracy [2] under multiple environments. Consequently, to accurately predict breeding values or phenotypic values is a challenge of primordial interest in GS, as the goal is to increase genetic gain. For this reason, when the traits of interest do not have a complex genetic architecture, this achievement is usually simple to accomplish. However, for complex heritable traits, traits with complex genetic architecture (such as grain yield) and with strong epistatic effects, this goal has limited success [3,4]. 

Reproducing Kernel Hilbert Spaces (RKHS) regression is a popular method in plant and animal breeding [5,6] for the prediction of complex traits and modeling complex interactions more efficiently. The central idea of an RKHS regression is to project the given original input data available in a finite dimensional space onto an infinite dimensional Hilbert space. Kernel methods can incorporate any statistical machine learning algorithm to the resulting transformed data, after using a kernel function. Empirical experience indicates that generally better results are accomplished with the transformed input. For this reason, RKHS methods are becoming more popular for analyzing nonlinear patterns in datasets collected in plant and animal breeding.

RKHS methods are very attractive because, in addition to being efficient for capturing nonlinear patterns, they are also efficient for data compression, as the transformed input has less dimensionally than the original input; this is to say, when the input is a matrix of dimensions n×p, with p≫n, the transformed input has a dimension of order n×n, which has less dimension and can reduce the computational complexity required during the training process. There are many transformations (kernel functions) used to capture nonlinear patterns in the original input, and each type of transformation is specialized for capturing some type of nonlinear pattern. However, it is impossible to capture all patterns with conventional linear statistical methods [5,6].

It should be noted that RKHS methods are not limited to regression as they are also powerful in the context of classification problems, where they are also efficient and popular. Support vector machine (SVM), which was proposed to the computer science community in the 1990s by Vapnik [7], is one of the most popular methods for classification based on kernels. 

In the context of GS, RKHS methods are increasingly accepted as rising evidence aids in increasing the accuracy of predictions using linear methods. For example, in a study about body weight of broiler chickens, Long et al. [8] reported a better prediction accuracy of RKHS methods over linear models. Crossa et al. [9] and Cuevas et al. [10] in wheat and maize found that the RKHS methods outperformed the linear methods. However, some authors have also reported minimal differences between RKHS methods and linear models [11,12], which is expected when the nonlinear patterns in the data are minimal or non-existing.

Moreover, empirical evidence has shown that MT models are more efficient than single-trait (ST) models [13]. Some reasons why MT models are chosen over ST models [14] are that: (1) they capture complex relationships between correlated traits in a more efficient way, (2) they take advantage of the degree of correlation between lines and traits, (3) MT models offer better interpretability than ST models, (4) they are computationally more parsimonious to train than ST models, (5) more precise estimates of random effects of lines and genetic correlations between traits are obtained, which allows for improvement of the index selection, (6) they become more efficient for indirect selection as the precision of genetic correlation parameter estimates increases, and (7) they improve hypothesis testing because they reduce type I and II errors [2] due to a more precise estimates of parameters.

However, the prediction performance of RKHS methods over conventional linear models, is not improved when a proper tuning process is not achieved. For example, when the Gaussian kernel is implemented, the bandwidth hyperparameter is set to the median of the average distances or to 1, which in some cases is not optimal and can cause the resulting prediction performance to be worse than conventional linear models. This implies that when a nonlinear kernel is performed in a model, an additional process is required to select the optimal hyperparameters to increase the prediction accuracy. However, it is also true that certain default hyperparameters frequently do an acceptable prediction performance but are not optimal. For this reason, to acquire the full power of any statistical machine learning method, a careful fine-tuning process should always be carried out.

Based on the above-mentioned considerations, we do a benchmarking study in this paper to compare the prediction performance implementing a multi-trait Gaussian kernel in the context of genomic prediction using the multivariate Bayesian Genomic Best Linear Unbiased Predictor (GBLUP) model. Since the Gaussian kernel only depends on one hyperparameter, only this hyperparameter was tuned under the following three strategies: (1) no tuning setting to the bandwidth parameter, (2) tuning using the grid search method, and (3) tuning using the Bayesian optimization method. This benchmark was carried out using 5 real datasets collected in real plant breeding programs.

## 2. Material and Methods

### 2.1. Dataset 1. Japonica

This dataset contains information on the phenotypic performance of four traits (GY = Grain Yield, PHR = Percentage of Head Rice Recovery, GC = percentage of Chalky Grain, PH = Plant Height) of rice as reported by Monteverde et al. [15] and evaluated over the course of five years (2009, 2010, 2011, 2012 and 2013). The genotypes evaluated were 93, 292, 316, 316 and 134 lines for years 2009, 2010, 2011, 2012 and 2013, respectively. This dataset contains 54 environmental covariates but, in this application, these covariates were not included in the analysis. In this dataset, a total of 1051 assessments were evaluated over five years. In this dataset, the genotypes evaluated were 320 and for each 44,598 markers remained after quality control that were coded with 0, 1 and 2. For more details about the data, see Monteverde et al. [15].

### 2.2. Dataset 2. Indica

This dataset contains information on the same traits as the Japonica dataset [15], with only three environments (years 2010, 2011 and 2012). In each year (environment), 327 genotypes were evaluated. Although this dataset contained environmental covariates they were not used in this study. The total number of observations in this balanced dataset was 981 since each line was included once in each environment. The genotyping-by-sequencing (GBS) markers datasets were filtered to retain markers with 50% missing data after imputation and a minor allele frequency (MAF) > 0.05. The markers remaining after quality control were 92,430 SNPs for each line and were coded as 0, 1 and 2, where 0 was used if the SNP was homozygous for the major allele, 1 if the SNP was heterozygous and 2 if the SNP was homozygous for the other allele. For more details about the data, see Monteverde et al. [15].

### 2.3. Dataset 3. Groundnut

This dataset was reported by Pandey et al. [16] with genotypic and phenotypic information for 318 genotypes and four environments. The traits measured were seed yield per plant (SYPP), pods per plant (NPP), pod yield per plant (PYPP) and yield per hectare (YPH). The environments were identified as: Aliyarnagar_Rainy 2015 (ENV1), Jalgoan_Rainy 2015 (ENV2), ICRISAT_Rainy 2015 (ENV3), and ICRISAT Post-Rainy 2015 (ENV4).

This dataset contained a total of 1272 observations and is balanced, since each genotype was included once in each environment. For each genotype, 8268 single nucleotide polymorphism (SNP, or SNPs in plural) markers (coded with 0, 1 and 2) were available after quality control. For more details about the data, see Pandey et al. [16].

### 2.4. Dataset 4. Cotton

This dataset was proposed by Gapare et al. [17] with genotypic and phenotypic information for 859 genotypes and seven environments [Myall Vale (MV), Collarenebri (CO), Bourke (BK), Emerald (EM), St. George (SG), Breeza (BR), Darling Downs (DD)]. The traits analyzed for the study were fiber length and strength.

This dataset contains a total of 859 observations and is not balanced, since each genotype was not included in each environment. For each genotype, 5000 single nucleotide polymorphism (SNP, or SNPs in plural) markers (coded with 0, 1 and 2) were available after quality control. For more details about the data, see Gapare et al. [17].

### 2.5. Dataset 5. Disease

This dataset contains 438 wheat genotypes (lines), three traits. PTR that denotes *Pyrenophora tritici-repentis* (PTR), SN denotes *Parastagonospora nodorum*, a major fungal pathogen of wheat fungal taxon, and SB that denotes *Bipolaris sorokiniana* (SB), that causes seedling diseases, common root rot and spot blotch of several crops such as barley and wheat. These 438 lines were evaluated in the greenhouse for six replicates during a period of time. The replicates were considered as environments (Env1, Env2, Env3, Env4, Env5, and Env6).

For the three traits evaluated, the total number of observations was 438 × 6 = 2628. 

DNA samples were genotyped using 67, 436 SNPs. For each marker, the genotype for each line was coded as the number of copies of a designated marker-specific allele carried by the line (absence = zero and presence = one). SNP markers with unexpected heterozygous genotypes were recoded as either AA or BB. Those markers that had more than 15% missing values or with MAF < 0.05 were removed. A total of 11,617 SNPs were still available for analysis after quality control and imputation.

### 2.6. Multi-Trait Kernel Model

This model is given in Equation (1) as:(1)$$Y=1_{n}\mu^{T}+X_{E}\beta_{E}+Z_{L}g+Z_{EL}gE+\epsilon$$
where Y is the matrix of phenotypic response variables of order $n\times n_{T}$ and ordered first by environments and then by lines, $n_{T}$ denotes the number of traits, $1_{n}$ is a vector of ones of length n, $\mu^{T}$ is a vector of intercepts for each trait of length $n_{T}$, T denotes the transpose of a vector or matrix, that is, $\mu ={\left[\mu_{1},\ldots ,\mu_{n_{T}}\right]}^{T},X_{E}$ is the design matrix of environments of order n×I, I denotes the number of environments, $\beta_{E}$ is the matrix of coefficients for environments with a dimension of $I\times n_{T}$, $Z_{L}$ is the design matrix of lines of order n×J, J denotes the number of lines, g is the matrix of random effects of lines of order $J\times n_{T}$ distributed as $g∼MN_{J\times n_{T}}\left(0,K,\Sigma_{T}\right)$, that is, with a matrix-variate normal distribution with parameters M=0, U=G and $V=\Sigma_{T}$, K is the Gaussian kernel (GK) that mimics a covariance matrix to capture the degree of similarity between lines, such as the genomic relationship matrix (Linear kernel) proposed by [18] that was built with marker data of order J×J and $\Sigma_{T}$ is the variance-covariance matrix of traits of order $n_{T}\times n_{T}$. $Z_{EL}$ is the design matrix of the genotype × environment interaction of order n×JI, gE is the matrix of genotype × environment interaction random effects distributed as $gE∼MN_{JI\times n_{T}}\left(0,Z_{E}Z_{E}^{T}⊙Z_{g}KZ_{g}^{T},\Sigma_{T}\right)$, where $\Sigma_{E}$ is a diagonal variance-covariance matrix of environments of order I×I and ⊙ denotes the Hadamard product. ϵ is the residual matrix of dimension $n\times n_{T}$ distributed as $\epsilon ∼MN_{n\times n_{T}}\left(0,I_{IJ},R\right)$, where R is the residual variance-covariance matrix of order $n_{T}\times n_{T}$.

The GK was computed using the GK function: (2)$$K\left(x_{i},x_{j}\right)=e^{-\gamma ∥x_{i}-x_{j}{∥}^{2}},\;\mathrm{with}\;\gamma >0$$
where $x_{i}$ and $x_{j}$ are the marker vectors for the ith and jth individuals (genotypes), respectively [19,20]. It is necessary to point out that the GK function was reparametrized (Caamal-Pat, et al. [1]) as:(3)$$K\left(x_{i},x_{j}\right)=e^{log\rho ∥x_{i}-x_{j}{∥}^{2}}\mathrm{with}\;\rho \in \left(0,1\right)$$
using the variable change ($\rho =e^{-\gamma }$). Subsequently, the three strategies for tuning the bandwidth (γ) hyperparameter used in this implementation are listed as follows:

(1) Manual tuning (no tuning, denoted as NT) setting the value of γ=1, which is equivalent to setting $\rho =e^{-1}$.

(2) Tuning using a grid search (GrS) strategy with 26 values in the grid for the values of ρ between 0.01 and 0.999 with increments of 0.04, this means that 26 values of ρ were evaluated. The average of the normalized root mean square error of each predicted trait ($NRMSE=\frac{1}{T}\overset{n}{\underset{t=1}{\sum }}NRMSE_{t}$, t=1,⋯,T) was used as metric for choosing the optimal ρ value in the inner testing set. 

(3) Tuning ρ using the Bayesian optimization (BO) method. The average NRMSE was also used as metrics to select the optimal ρ value in the inner testing set. 

The implementation of this model with the three strategies of tuning the ρ hyperparameter of the GK was carried out in the R statistical software [21,22].

### 2.7. Evaluation of Prediction Performance

In each of the five datasets, the seven outer fold cross validation was implemented (Montesinos-López et al. [19]). For this reason, 7−1 folds were assigned to the outer-training set, while the remaining were assigned to the outer-testing set until each of the 7 folds were tested once. For tuning the bandwidth hyperparameter of the Gaussian kernel five nested cross-validations was used; that is to say, the outer-training was divided into five groups where four were used for the inner training set (80% of the training) and one for the validation (inner-testing) set (20% of the outer training). Next, the average of the five validation folds was reported as the metric of prediction performance to select the optimal hyperparameter (bandwidth of the Gaussian kernel). Using this optimal hyperparameter (band width), the multi-trait kernel model (1) was then refitted with the whole outer-training set (the 7−1 folds), and finally, the prediction of each outer-testing set was obtained. 

The prediction accuracy was reported in terms of the average normalized root mean square error for each trait ($NRMSE_{t}=\frac{1}{7}\sum_{k=1}^{7}NRMSE_{t,k}=\frac{1}{7}\sum_{k=1}^{7}\frac{RMSE_{t,k}}{{\bar{y}}_{t,k}}$, for $t=1,\cdots ,n_{T}$, where $n_{T}$ is the number of predicted traits; $RMSE_{t,k}$ and ${\bar{y}}_{t,k}$ denote the NRMSE and the mean of the t-th trait for the kth fold, respectively), where $RMSE_{t,k}=\sqrt{\frac{1}{n_{k}}(\overset{n_{k}}{\underset{i=1}{\sum }}{\left(y_{it}-\hat{f_{t}}\left(x_{i}\right)\right)}^{2}}$ denoting the root mean square error of the t-th trait for the kth fold. In addition to the NRMSE for each trait, we also reported the average NRMSE of all traits as follows: $NRMSE=\frac{1}{T}\sum_{t=1}^{T}NRMSE_{t}$. These metrics were computed under the three strategies for tuning the bandwidth $\left(\gamma \right)$ hyperparameter used in the implementation of the model, so that $NRMSE_{NT}$, $NRMSE_{GrS}$ and $NRMSE_{BO}$ denote the NRMSE of no tuning (NT), grid search (GrS) and Bayesian optimization (BO) tuning strategies, respectively. The relative efficiencies were also reported and that were computed as: $$RE_{GrS}=\frac{NRMSE_{NT}}{NRMSE_{GrS}}$$
$$RE_{BO}=\frac{NRMSE_{NT}}{NRMSE_{BO}}$$

When $RE_{GrS}>1$ ($RE_{BO}>1$), the best performance prediction in terms of NRMSE was obtained using the GrS (BO) strategy, when $RE_{GrS}<1$ ($RE_{BO}<1$), the NT strategy was superior in terms of prediction accuracy and when $RE_{GrS}=1$ ($RE_{BO}=1$), both strategies of hyperparameter tuning were equally efficient. We also computed the relative efficiency in terms of NRMSE between the grid search strategy (GrS) and Bayesian optimization (BO) strategy ($RE_{GrS/BO}=NRMSE_{GrS}/NRMSE_{BO}$) and the interpretation is the same as the previous example. 

## 3. Results

The results are provided in three sections for Japonica, Indica and Groundnut datasets 1–3, respectively. The results from dataset 1 (Japonica) are given in Table A1, Figure 1A–D, and Appendix C
Table A4. The results from dataset 2 (Indica) are in Table A2, Figure 2A–D, and Appendix C
Table A5. The results from dataset 3 (Groundnut) are shown in Table A3, Figure 3A–D, and Appendix C
Table A6.

The results from dataset 4 (Cotton) can be found in Appendix A, whereas results from dataset 5 (Diseases) are found in Appendix A. 

### 3.1. Dataset 1 Japonica

The prediction performance for each environment and across environments (Global) at Japonica’s dataset in terms of normalized root mean squared error (NRMSE) and relative efficiency (RE) comparing the three methods of hyperparameter tuning (no tuning, grid search (GrS), and Bayesian optimization), under the 7FCV strategy are provided. NRMSE_GC, NRMSE_GY, NRMSE_PH and NRMSE_PHR denote the NRMSE of traits CC, GY, PH and PHR. 

As can be seen in Figure 1 and Table A1, in terms of NRMSE for the GC trait the best performance under the GrS strategy was in environments 2009 (1.124), 2010 (0.913), 2012 (0.887), and 2013 (0.730), while the environments with best RNMSE under the BO strategy were 2011 (0.774), 2012 (0.887), and Global (0.404). For trait GY, the best RMSE values were observed under the BO strategy in environments 2010 (0.818), 2011(0.875), 2012 (0.858), 2013(0.865) and global (0.491). The exception was in the 2009 environment where the best NRMSE value was 0.984 under the NT strategy. 

For the PH variable, the best predictions (lower NRMSE) were observed under the BO strategy [2010 (0.639), 2011 (0.757), 2013 (0.664) and Global (0.425)]. In the 2009 environment, the lowest NRMSE was 0.695 under the NT tuning strategy, while in the 2012 environment, the lowest NRMSE=0.653 was observed under the GrS strategy. For trait PHR, the best performance in terms of NRMSE of most environments was observed under the BO strategy [2009 (0.838), 2011 (0.811), 2012 (0.925), 2013 (0.837) and global (0.532)]. The year 2010 was an exception, as the best NRMSE was 0.827 and was found under the GrS strategy (see Figure 1 and Table A1). The standard error of prediction performance for every environment and across environments (Global) is provided in Appendix C
Table A4.

Across traits, the prediction performance can be observed in Figure 1A (Table A1), were the best predictions (lower NRMS) were observed under the BO and GrS strategies and the worst under the NT strategy. In addition, across traits we can observe that the RE of comparing the NT strategy versus BO strategy for each environment and across environments were 1.060 (2009), 1.0849 (2010), 1.028 (2011), 1.03 (2012), 1.0427 (2013) and 1.031 (Global) (Figure 1B; Table A1). This indicates that the BO method outperformed NT strategy in terms of NRMSE in all the environments mentioned by 6.03% (2009), 8.49% (2010), 2.83% (2011), 3% (2012), 4.27% (2013), and 3.1% (Global) (Figure 1B; Table A1), respectively. While the RE of comparing the NT strategy versus GrS strategy for each environment and across environments were 1.0494 (2009), 1.0889 (2010), 1.0193 (2011), 1.0261 (2012), 1.0284 (2013) and 1.0212 (Global) (Figure 1B; Table A1). This result indicates that the GrS method outperformed NT strategy in terms of NRMSE in all the environments mentioned by 4.94% (2009), 8.89% (2010), 1.93% (2011), 2.61% (2012), 2.84% (2013), and 2.12% (Global). Finally, the RE of comparing the GrS method versus the BO method were 1.0104 (2009), 0.9963 (2010), 1.0008 (2011), 1.004 (2012), 1.013 (2013) and 1.009 (Global) (Figure 1B; Table A1). This means that the BO strategy is slightly better in terms of prediction performance than the GrS method since the RE were slightly superior to one.

In Figure 1C (Table A1), the prediction performance is provided for each trait across environments, while in Figure 1D the relative efficiencies of comparing NT versus BO, NT versus GrS, and GrS and BO, are provided and show that in all traits, the best strategies for tuning in terms of NRMSE are the BO and GrS method without relevant differences between the GrS and BO method. 

### 3.2. Dataset 2 Indica

NRMSE_GC, NRMSE_GY, NRMSE_PH and NRMSE_PHR denote the NRMSE of traits CC, GY, PH, and PHR, respectively. Figure 2 and Table A2 shows that in terms of RMSE for the GC trait, the best performance under the GrS strategy was in environments 2010 (0.918), 2011 (0.92), 2012 (0.943), and Global (0.924). For trait GY the best RMSE values were observed under the GrS strategy in environments 2010 (0.915), 2011(0.825), and Global (0.716). However, in 2012, the best NRMSE value was 0.984 under the NT strategy.

For PH trait, the best predictions (lower NRMSE) were observed under the GrS strategy [2010 (0.422), 2012 (0.692), and Global (0.607)], with the exception in 2011 environment where the lowest NRMSE was 0.87 under the BO tuning strategy. For trait PHR the best performance in terms of NRMSE of most environments was also observed under the GrS strategy [2011 (0.866), 2012 (0.8), and global (0.8)], except in year 2010, where the best NRMSE was 0.819 using the BO strategy. Further details are given in Figure 2 and Table A2. The standard error of prediction performance for every environment (Global) is provided in Appendix C
Table A5.

As we summarize across traits for each environment, we can see in Figure 2A and Table A2, that the best predictions (lower NRMS) were observed under the BO and GrS strategies and the worst under the NT strategy. We can also observe that the RE of comparing the NT strategy versus BO strategy for each environment and across environments were 1.12 (2010), 1.056 (2011), 1.039 (2012), and 1.064 (Global) (Figure 2B and Table A2). This indicates that the BO method outperformed NT strategy in terms of NRMSE in all environments by 12% (2010), 5.6% (2011), 3.9% (2012), and 6.4% (Global). The RE of comparing the NT strategy versus GrS strategy for each environment and across environments were 1.129 (2010), 1.061 (2011), 1.046 (2012), and 1.068 (Global) (Figure 2B and Table A2). This indicates that the GrS method outperformed the NT strategy in terms of NRMSE in all the environments by 12.9% (2010), 6.1% (2011), 4.6% (2012), and 6.8% (Global). Finally, the RE of comparing the GrS method versus the BO method were 0.991 (2010), 0.995 (2011), 0.993 (2012), and 0.996 (Global) (Figure 2B and Table A2). These results indicate that the BO strategy is slightly worse in terms of prediction performance than the GrS method since in most cases, the RE is less than one.

The prediction performance in terms of NRMSE of each trait across environments are given in Figure 2C and Table A2 and the relative efficiencies of comparing NT versus BO, NT versus GrS and GrS and BO are given in Figure 2D and Table A2, where in three out of four traits the best strategies for tuning are the BO and GrS, while the worst was the NT strategy. It should be noted that there are no relevant differences between the BO and GrS methods. 

### 3.3. Dataset 3 Groundnut

Here, NRMSE_NPP, NRMSE_PYPP, NRMSE_SYPP and NRMSE_YPH denote the NRMSE of traits NPP, PYPP, SYPP and YPH. As shown in Table A3, in terms of NRMSE for the NPP trait, the best performance under the GrS strategy was in environments ALIYARNAGAR_R15 (0.808), ICRISAT_R15 (0.786) and Global (0.77), while the environments with the best RNMSE under the BO strategy were ICRISAT_PR15-16 (0.902) and JALGOAN_R15 (0.808). For trait PYPP, the best NRMSE values were observed under the BO strategy in ICRISAT_PR15-16 (0.954), ICRISAT_R15 (0.772) and JALGOAN_R15 (0.836). ALIYARNAGAR_R15 and Global were the exception where the best NRMSE values were 0.934 and 0.782 under the GrS strategy (Figure 3)

In Figure 3 and Table A3 we can also see that in terms of NRMSE for the SYPP trait the best performance under the GrS strategy was in environments ALIYARNAGAR_R15 (0.933), ICRISAT_PR15-16 (0.944), ICRISAT_R15 (0.787) and Global (0.792), while the environment with the best RNMSE under BO strategy was JALGOAN_R15 (0.838). For trait YPH the best NRMSE values were observed under the GrS strategy in environments ALIYARNAGAR_R15 (0.811), ICRISAT_PR15-16 (0.915), JALGOAN_R15 (0.767) and Global (0.784). In the case of environment ICRISAT_R15, the best NRMSE value was 0.67 under the BO strategy. More details are provided in Table A3. The standard error of prediction performance for every environment and across environments (Global) are provided in the Appendix C
Table A6.

Summarizing across traits for each environment, the best predictions (lower NRMS) were observed under the BO and GrS strategies in most cases, while the worst were under the NT strategy (Figure 3A; Table A3). Across traits we can also observe that the RE of comparing the NT strategy versus the BO strategy from each environment and across environments were 1.013 (ALIYARNAGAR_R15), 1.045 (ICRISAT_R15), 1.068 (ICRISAT_PR15-16), 1.042 (JALGOAN_R15) and 1.044 (Global) (Figure 3B; Table A3). This indicates that the BO method outperformed the NT strategy in terms of NRMSE in all environments by 1.3% (ALIYARNAGAR_R15), 4.5% (ICRISAT_R15), 6.8% (ICRISAT_PR15-16), 4.2% (JALGOAN_R15) and 4.4% (Global). Meanwhile, the RE of comparing the NT strategy versus GrS strategy for each environment and across environments were 1.026 (ALIYARNAGAR_R15), 1.043 (ICRISAT_R15), 1.07 (ICRISAT_PR15-16), 1.04 (JALGOAN_R15) and 1.046 (Global) (Figure 3B; Table A3). This indicates that the GrS method outperformed the NT strategy in terms of NRMSE in all environments by 2.6% (ALIYARNAGAR_R15), 4.3% (ICRISAT_R15), 7% (ICRISAT_PR15-16), 4% (JALGOAN_R15) and 4.6% (Global). Finally, the RE of comparing the GrS method versus the BO method were 0.987 (ALIYARNAGAR_R15), 0.997 (ICRISAT_R15), 1.002 (ICRISAT_PR15-16), 1.001 (JALGOAN_R15) and 0.997 (Global) (Figure 3B). This means that the BO strategy is slightly worse in terms of prediction performance than the GrS method since the RE in most of the cases was less than one. For more details, see Table A3.

The prediction performance in terms of NRMSE of each trait across environments is given in Figure 3C and Table A3, while the relative efficiencies of comparing NT versus BO, NT versus GrS and GrS and BO are given in Figure 3D and Table A3, where we can appreciate that in the four’s traits evaluated, the best strategies for tuning were the BO and GrS, while the worst was the NT strategy. No relevant differences were observed between the BO and GrS methods.

## 4. Discussion

As a predictive methodology, GS can help increase genetic gain by saving significant resources since candidate phenotypes do not need to be measured in the field, as they are predicted [23]. However, a number of factors still need to be improved for prediction performance to be optimized. One of these factors is the choice of the statistical machine learning that will be used. In this regard, there are statistical machine learning methods that can only capture linear patterns, while others are able to capture non-linear patterns [19]. 

Kernel methods are very attractive since they are able to capture non-linear patterns and are very versatile, as they can be used with many statistical machine learning methods. For example, kernel methods can be applied in conventional mixed models, in support vector machines and even in many others machine learning methods such as random forest, and gradient boosting machine providing a modified input. In conventional mixed models, the use of kernels is quite straightforward since the genomic relationship matrix that is provided in this case is replaced by a particular kernel, enhancing the power of mixed models to capture nonlinear patterns in the data [19]. However, many kernels such as the Gaussian kernel implemented in this study have hyper-parameters that must be appropriately tuned to guarantee successful implementation.

For this reason, in this study we evaluated the influence in terms of prediction performance of three tuning strategies (manual tuning, grid Search and Bayesian optimization) under a multi-trait Bayesian GBLUP model. We found that using the grid search and Bayesian optimization outperform the prediction accuracy of the manual tuning by 2.1% and 3.1%, respectively, in the japonica dataset, by 6.4% and 6.8%, respectively, in the Indica dataset, by 4.4% and 4.6%, respectively, in the groundnut dataset, by 1.9% and 2.1%, respectively, in the Cotton dataset, and by 2.3% and 2.7%, respectively, in the disease dataset. 

About the time for implementing the tunning methods we found that the grid Search method required around 15% more time for its implementation than the Bayesian optimization method, and around 20 times more expensive in computation resources than the manual tuning. The tuning process is more expensive in terms of computational resources. The grid search approach was slightly more costly than the Bayesian optimization since the size of the grid contain 26 values, however the larger the number of values in the grid search the larger the computational resources required by the grid search.

We found differences in prediction performance in each environment, and larger differences were observed when the environments represented years, since many times the year-to-year variability is significant. For example, in dataset 1 (Japonica) we observed higher prediction error in year 2009, compared to the other years and this can be attributed mostly to the effect of years and less to the unbalance in the number of genotypes evaluated in each environment. In addition, relevant differences in terms of prediction performance between environments were found in dataset 3 (Groundnut) where environments ALIYARNAGAR_R15 and ICRISAT_PR15-16 resulted in the worst prediction performance and environments ICRISAT_R15 and JALGOAN_R15 with the best predictions. These results point out that even in the same year the location-to-location variability is considerably high. 

In addition, for some datasets we found significant differences in terms of prediction performance between traits. For example, in dataset 1 (Japonica) the best predictions were observed in traits GC and PH and the worst in traits GY and PHR. While in dataset 2 (Indica) the best predictions were observed in traits GY and PH, while the worst in traits GC and PHR. In dataset 3 (Groundnut) the four traits showed a similar prediction performance.

The improvement in prediction accuracy of the grid search and Bayesian optimization is not very significant compared with the manual tuning; however, this is data dependent. By data dependent we mean that, if the dataset contain complex non-linear patterns, then using kernels with the appropriate implementation and tuned methods will result in important improvement in prediction accuracies with respect to not using kernels. However, if the data only contain linear patterns, we cannot expect an improvement in prediction accuracy using non-linear kernels. We also need to be aware that we are under a multi-trait framework, which lends to greater difficulty in selecting the bandwidth hyperparameter that can work simultaneously for all traits under study. In the context of tuning the bandwidth for Gaussian kernels, a greater gain in prediction accuracy was observed using grid search and Bayesian optimization as was shown by Montesinos-López et al. [20].

## 5. Conclusions

In this study we showed the importance of carefully tuning the Gaussian kernel to improve the prediction accuracy. We found that we can increase the prediction accuracy between 1.9% and 6.8% by tuning the Gaussian kernel using Bayesian optimization or the grid search method. We did not find any relevant differences between tuning with Bayesian optimization and the grid search method. In general, the results indicated that modest gain in prediction accuracy were obtained for some datasets, while in others major improvements were achieved. We encourage researchers dedicating sufficient time for the tuning process. It is also important to point out that the degree of improvement in prediction accuracy can be influenced by the metric used for evaluating the prediction performance in the validation set and for this reason our results are not conclusive. We encourage future benchmark studies to be able to see the influence of the metric used.

## Figures and Tables

**Figure 1 genes-13-02279-f001:**
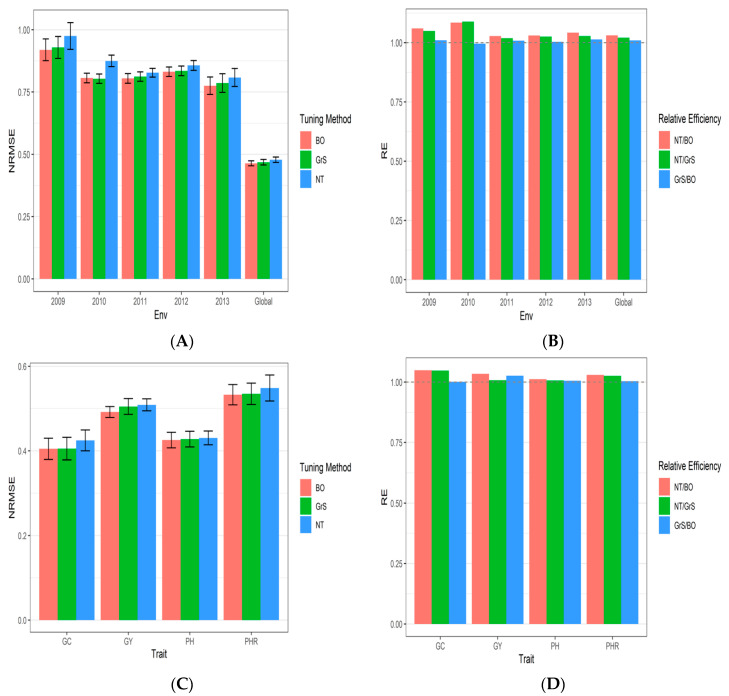
(**A**) The prediction performance for dataset 1, Japonica dataset in terms of normalized root mean squared error (NRMSE) for each year (2009–2013), across years (Global), and across traits with three strategies of tuning (BO, GrS and NT) under 7 Fold Cross-Validation (7FCV). (**B**) The relative efficiency for each environment (2009–2013) and across environments (Global) and across traits (CG, GY, PH and PHR) with three strategies of tuning (BO, GrS and NT) under 7 Fold Cross-Validation (7FCV). (**C**) The prediction performance in terms of normalized root mean squared error (NRMSE) for each trait (CG, GY, PH and PHR) across years with three strategies of tuning (BO, GrS and NT) under 7 Fold Cross-Validation (7FCV). (**D**) The relative efficiency for each trait (CG, GY, PH and PHR) across years with three strategies of tuning (BO, GrS and NT) under 7 Fold Cross-Validation (7FCV). When RE > 1 the denominator method outperforms the numerator in terms of prediction performance.

**Figure 2 genes-13-02279-f002:**
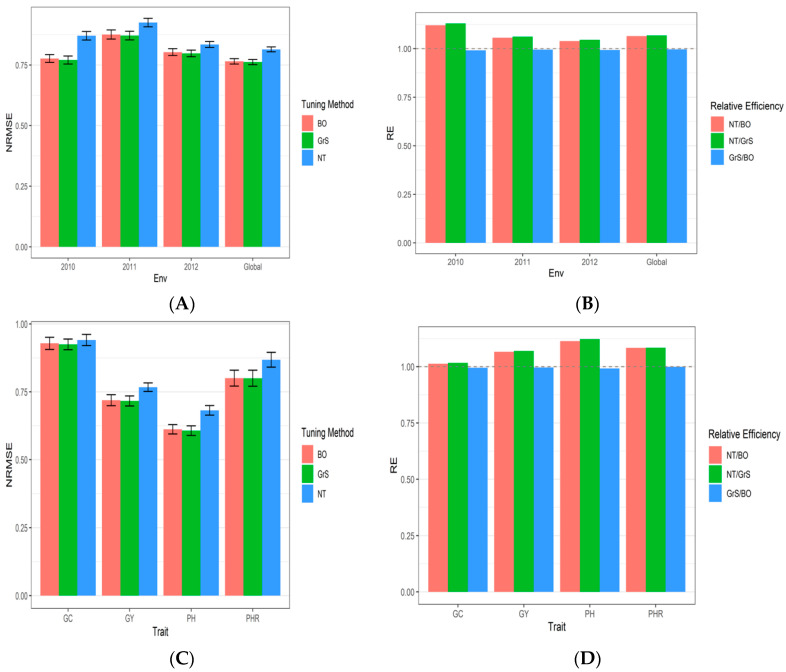
(**A**) Prediction performance for dataset 2, Indica dataset in terms of normalized root mean squared error (NRMSE) for each year (2010–2012) across traits (CG, GY, PH and PHR), across years and across traits (Global) with three strategies of tuning (BO, GrS and NT) under 7 Fold Cross-Validation (7FCV). (**B**) The relative efficiency for each environment (2010–2012) across traits and across environments and across traits (Global) with three strategies of tuning (BO, GrS and NT) under 7 Fold Cross-Validation (7FCV). (**C**) Prediction performance in terms of normalized root mean squared error (NRMSE) for each trait (CG, GY, PH and PHR) across years with three strategies of tuning (BO, GrS and NT) under 7 Fold Cross-Validation (7FCV). (**D**) The relative efficiency for each trait (CG, GY, PH and PHR) across environments with three strategies of tuning (BO, GrS and NT) under 7 Fold Cross-Validation (7FCV). When RE > 1 the denominator method outperforms the numerator in terms of prediction performance.

**Figure 3 genes-13-02279-f003:**
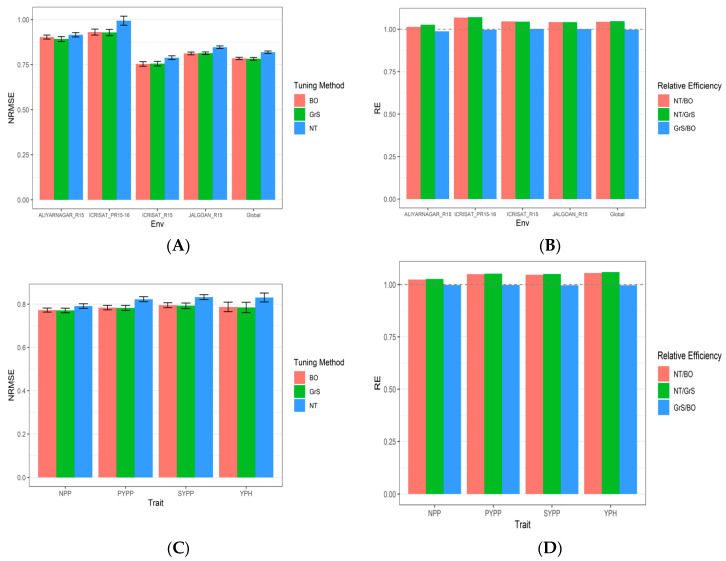
(**A**) The prediction performance for dataset 3, Groundnut dataset in terms of normalized root mean squared error (NRMSE) for each environment across traits (ALIYARNAGAR_R15, ICRISAT_PR15-16 ICRISAT_R15 and JALGOAN_R15), across environments and traits (Global) with three tuning strategies (BO, GrS and NT) under 7 Fold Cross-Validation (7FCV). (**B**) The relative efficiency for each environment across traits (ALIYARNAGAR_R15, ICRISAT_PR15-16 ICRISAT_R15 and JALGOAN_R15), across environments and traits (Global) with three tuning strategies (BO, GrS and NT) under 7 Fold Cross-Validation (7FCV). (**C**) The prediction performance in terms of normalized root mean squared error (NRMSE) for each trait (NPP, PYPP, SYPP, YPH)) across environments with three tuning strategies (BO, GrS and NT) under 7FCV. (**D**) The relative efficiency for each trait (NPP, PYPP, SYPP, YPH) across environments with three tuning strategies (BO, GrS and NT) under 7 Fold Cross-Validation (7FCV). When RE > 1, the denominator method outperforms the numerator in terms of prediction performance.

## Data Availability

Phenotypic and genomic data can be downloaded from the link: https://github.com/osval78/Multivariate_Tuning_Kernel_Method.

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
