# Peer review of "A Multi-Trait Gaussian Kernel Genomic Prediction Model under Three Tunning Strategies"

_genes, 2022, doi:10.3390/genes13122279_

Round 1

Reviewer 1 Report

A multi-trait Gaussian kernel genomic prediction model for three tunning strategies

The authors aimed to explore the tuning process under a multi-trait framework using the Gaussian kernel with a multi-trait Bayesian Best Linear Unbiased Predictor (GBLUP) model. They found that using Grid search and Bayesian optimization improves between 1.9 and 6.8% of the prediction accuracy regarding using manual tuning. While the improvement in prediction accuracy in some cases can be marginal, it is really important to carry out the tuning process carefully to improve the accuracy of the GS methodology—even though this entails greater computational resources are needed.

The method section was prepared perfectly.

This manuscript may need some minor revisions:

1. Line 83: Please describe how they reduce type I and II errors.

2. Line 113, 122: “coded with 0, 1 and 2” which one AA?

3. Line: 467: “improvements. were achieved.” Should be “improvements were achieved.” (delete the dot)

4. If you have genetic correlation matrixes could you compare them also?

Author Response

genes-2040773 A multi-trait Gaussian kernel genomic prediction model for three tunning strategies

REVIEWER 1

A multi-trait Gaussian kernel genomic prediction model for three tunning strategies

The authors aimed to explore the tuning process under a multi-trait framework using the Gaussian kernel with a multi-trait Bayesian Best Linear Unbiased Predictor (GBLUP) model. They found that using Grid search and Bayesian optimization improves between 1.9 and 6.8% of the prediction accuracy regarding using manual tuning. While the improvement in prediction accuracy in some cases can be marginal, it is really important to carry out the tuning process carefully to improve the accuracy of the GS methodology—even though this entails greater computational resources are needed.

 The method section was prepared perfectly.

        RESPONSE: Many thanks to the reviewers for the time. Very much appreciated

This manuscript may need some minor revisions:

  1. Line 83: Please describe how they reduce type I and II errors.

RESPONSE Correction done in the new version of the paper. See line 83.

  1. Line 113, 122: “coded with 0, 1 and 2” which one AA?

RESPONSE Correction on lines 123-126.

  1. Line: 467: “improvements. were achieved.” Should be “improvements were achieved.” (delete the dot)

 RESPONSE: Correction done in the new version of the paper. See lines 487-488.

  1. If you have genetic correlation matrices, could you compare them also?

        RESPONSE Correction done in the new version of the paper. We did not compute genetic correlation matrices.

Reviewer 2 Report

Comments to the Authors

Your research could be valuable for many. The content is generally good, but needs some improvement and tuning. Try to put key results precisely in the results section and avoid redundancy. If the Figures work, then put different key findings in the Table or vise-versa. There is greater potential to improve the discussion section. For example, in line 283, the performance of 2009 and 2013 was comparable, but they had 93 and 316 genotypes. Such inputs can be brought to the discussion. Similarly, from Figure 1 A) difference in performance in different years due to year or differences in the genotypes? Such discussion would increase the interest and acceptability of this manuscript. In lines 452 and 453, you mentioned that the prediction accuracy of Grid search and Bayesian optimization is data dependent, supporting this statement with the presented findings in this manuscript. Please consider adding information on the relative computational resources for respective tuning methods to allow the audience and scientists to choose their tuning methods. Please go through the comments and suggestions throughout the text and improve those.

Overall comments:

Add citations

Overall describe the data set more details.

Please check for typos.

Clarify the evaluation of prediction with seven-fold cross validation

Precise results, and avoid redundancy

Improve discussion

Thank you.

Author Response

genes-2040773 A multi-trait Gaussian kernel genomic prediction model for three tunning strategies

REVIEWER 2

Comments to the Authors

Your research could be valuable for many. The content is generally good, but needs some improvement and tuning. Try to put key results precisely in the results section and avoid redundancy. If the Figures work, then put different key findings in the Table or vise-versa.

        RESPONSE: Many thanks. Correction done in the new version of the paper. All Tables were sent to

                              Appendix.

There is greater potential to improve the discussion section. For example, in line 283, the performance of 2009 and 2013 was comparable, but they had 93 and 316 genotypes. Such inputs can be brought to the discussion.

Similarly, from Figure 1 A) difference in performance in different years due to year or differences in the genotypes? Such discussion would increase the interest and acceptability of this manuscript.

RESPONSE: Correction done. See lines 452-468.

In lines 452 and 453, you mentioned that the prediction accuracy of Grid search and Bayesian optimization is data dependent, supporting this statement with the presented findings in this manuscript.

RESPONSE: Correction done. See lines 470-475.

Please consider adding information on the relative computational resources for respective tuning methods to allow the audience and scientists to choose their tuning methods. Please go through the comments and suggestions throughout the text and improve those.

RESPONSE: Yes and thanks. See. See lines: 443-450.

Overall comments:

Add citations

RESPONSE:Citation added See lines: 612-615,639-640.

Overall describe the data set more details.

RESPONSE: Done. See lines: 113, 114, 123-126, 137,147.

Please check for typos.

RESPONSE: Typos corrected in the new version of the paper. See lines: 2, 4, 23, 24, 25, 29, 60, 64, 68, 69,86, 96, 100,117, 208, 213, 395,431, 433, 479.

Clarify the evaluation of prediction with seven-fold cross validation

RESPONSE: Correction done in the new version of the paper. See lines: 206, 208 and 213.

Precise results, and avoid redundancy

RESPONSE:; Thanks. We have corrected. See lines all tables in the text that were send to appendix.

Improve discussion

RESPONSE: Thanks. Correction done in the new version of the paper. See lines: 444-469.
